# Effects of Prism Adaptation on Reference Systems for Extrapersonal Space in Neglect Patients

**DOI:** 10.3390/brainsci9110327

**Published:** 2019-11-16

**Authors:** Laura Abbruzzese, Alessio Damora, Gabriella Antonucci, Pierluigi Zoccolotti, Mauro Mancuso

**Affiliations:** 1Tuscany Rehabilitation Clinic, Montevarchi, Piazza del Volontariato 2, 52025 Montevarchi, Arezzo, Italy; 2Department of Psychology, Sapienza University of Rome, Via dei Marsi 78, 00185 Rome, Italy; 3Neuropsychology Centre, IRCCS Santa Lucia Foundation, Via Ardeatina 306/354, 00142 Rome, Italy; 4Physical and Rehabilitative Medicine Unit, NHS-USL Tuscany South-Est, Via Senese 169, 58100 Grosseto, Italy

**Keywords:** egocentric neglect, allocentric neglect, prism adaptation

## Abstract

Up to now, rehabilitation of unilateral spatial neglect has focused on egocentric forms of neglect, whereas less is known about the possibility to improve allocentric deficits. The present study aimed to examine the efficacy of prism adaptation (PA) training on patients with different forms of neglect: egocentric, allocentric, or mixed. Twenty-eight patients were assessed with specific neglect tests before (T0) and after (T1) 10 sessions of PA training. Performance in the Apples Cancellation test was used to identify patients with egocentric (*n* = 6), allocentric (*n* = 5), or mixed (*n* = 17) forms of neglect. In the overall group of patients, PA training produced significant improvements in performance across different neglect tests. In terms of the egocentric–allocentric distinction, the training was effective in reducing omissions in the left part of space in the Apples Cancellation test both for patients with egocentric neglect and mixed neglect. By contrast, errors of commissions (marking the inability to detect the left part of the target stimulus, i.e., allocentric neglect) remained unchanged after PA in patients with allocentric neglect and actually increased marginally in patients with mixed neglect. The PA training is effective in improving egocentric neglect, while it is ineffective on the allocentric form of the disturbance. Notably, the allocentric component of neglect is frequently impaired, although this is most often in conjunction with the egocentric impairment, yielding the mixed form of neglect. This stresses the importance of developing exercises tuned to improving allocentric neglect.

## 1. Introduction

Unilateral spatial neglect is defined as a deficit in responding, orienting, or initiating action toward contralesional stimuli [1]. Neglect affects perception and mental representation of spatial information, as well as planning and execution of motor actions. This disorder disrupts essential functions of daily life, affecting late functional outcome [2,3], prolonging inpatient rehabilitation facility stay [2,3,4], increasing the risk of falls [3,5], and decreasing the likelihood of home return even after completion of intensive rehabilitation programs [2,3,6].

Currently, there is consensus for the role of at least two spatial reference frames in neglect: an egocentric reference frame based on body-centered coordinates, related to grasping and manipulating objects, and an object-centered reference frame, which codes spatial information in relation to the perceived object independently of body position. Thus, in the cases of right-hemispheric lesions, some patients show errors on their contralesional side (egocentric neglect), such as omissions of words or other target stimuli on the left side of a sheet of paper [7], whereas other patients show errors on the left side of every object (allocentric neglect) regardless of its position in space [7,8,9,10,11].

Although a good deal of information is available on these two different forms of neglect, rehabilitation has commonly dealt with this syndrome without considering the different prognoses [12] and sensibility to treatment [13]. Indeed, very little is known about the possibility of ameliorating the specific (egocentric and allocentric) components of neglect, even if previous evidence has already underscored the importance of actively intervening on both components of the disorder [14]. In fact, patients with object-centered neglect or mixed forms of neglect have lower Barthel Index scores and greater difficulties in everyday activities when compared to patients with egocentric neglect [15]. Nevertheless, current understanding of the clinical impact of allocentric disturbance appears incomplete [16].

Among the rehabilitative treatments for neglect, prism adaptation (PA) has proven to be a promising training procedure, with generally positive outcomes (see [17,18,19,20,21], for reviews). Furthermore, from a clinical standpoint, PA is a well-defined procedure that can be effectively standardized.

One study assessed the effects of a short period of PA treatment on post-acute patients (right hemispheric stroke and severe neglect) with mixed forms of neglect, showing an improvement of egocentric neglect in the absence of substantial changes in the allocentric components of the disturbance, i.e., commission errors in crossing out stimuli with a left-sided opening [13]. These errors actually increased, although not significantly, in parallel with the decrease of egocentric neglect. The authors attributed this phenomenon to damage of the dorsal way and to lesions in the post-central cortical areas correlated with the allocentric frame of reference. Therefore, this study raised the possibility that PA has differential effects on egocentric and allocentric reference frames. However, the short treatment (with only four sessions of PA) and the small number of patients with varying degrees of impairment limited the significance of the conclusions. Some previous studies [21,22] yielded similar results in comparing the short-term experimental effects of PA on chimerical faces and objects, arguing that performance in such tasks cannot be improved by wearing prism glasses. These data are consistent with the notion that PA influences dorsal stream processes without influencing perceptual biases in the ventral stream in patients with neglect [21]. Thus, while the exploratory eye movements of a patient with neglect were clearly shifted toward the left after PA, he still showed no awareness of the left side of the stimuli he was actively exploring [22].

Along similar lines, we reported the differential effect of PA training on egocentric and allocentric reference frames in a patient with severe egocentric and allocentric neglect (using a training protocol developed by Farnè et al. [23]). The PA training selectively improved egocentric neglect but also produced an increase in commission (allocentric) errors as the patient enlarged her exploration of left space [14]. 

Overall, current evidence indicates an effect of PA training on egocentric neglect but less or no effect on allocentric neglect. However, evidence is still limited and, critically, only patients with mixed forms of neglect were examined up to now. The present study aimed to examine the efficacy of the PA training on patients with either egocentric, allocentric, or mixed forms of neglect.

## 2. Materials and Methods

### 2.1. Participants

All patients were admitted to the hospital section of neurological rehabilitation of Grosseto for rehabilitative treatment. To be included in this study, patients were selected for the presence of neglect after brain damage. The inclusion criteria were as follow:Presence of a right cerebral lesion diagnosed by a neurological evaluation;Presence of motor and functional signs of neglect and pathological performances in at least two tests for neglect (Behavioural Inattention Test (BIT) conventional subtests and Bells Test, as reported in Table 1).

Patients with hemianopia and previous neurological and/or psychiatric pathologies were excluded. Exclusion of patients with hemianopia was due to the documented absence, in the presence of visual-field deficit, of the low-order visuomotor reorganization induced by PA, which promotes a resetting of the oculomotor system, leading to an improvement of high-order visuospatial representations [24].

Overall, we enrolled 28 patients (18 males and 10 women; aged between 43 and 85 (mean = 66.3, SD = 13.1). Based on their performance of the Apples test (see below), six patients showed egocentric neglect, five allocentric neglect, and seventeen a mixed form of neglect (Table 1).

The study was approved by the Local Ethics Committee of Tuscany Area Vasta Sud-Est on 16 June 2014 (project identification code: usl9-2014-016-NEGLECT), and conducted in accordance with the Declaration of Helsinki. Patients gave their written informed consent to participate in the study.

### 2.2. Design and Procedure

Patients were examined at T_0_ (baseline evaluation, before the rehabilitative treatment) and at T_1_ (follow-up, after the treatment). All patients underwent standardized testing in order to assess their exploration skills (see below). Neuropsychological evaluations were performed by a certified neuropsychologist. The treatment was administered by trained therapists. 

#### 2.2.1. Neglect Evaluation

The *Apples Cancellation Test* [15] is a test that aims to detect both egocentric and allocentric forms of neglect. It consists of a cancellation task in which 150 apple symbols are shown in a pseudo-randomly scattered way over an A4 sheet. Fifty apples are complete (targets), while the others are incomplete on the left or right side (distractors). Targets and distractors are evenly distributed. For the purpose of analysis, the page is divided into a grid (not visible to the patient) of five columns. Boxes were numbered from 1 to 10. Boxes 1 and 2 appear on the extreme left, and boxes 9 and 10 on the extreme right side of the sheet. The patient is given 5 min to cancel out all the complete apples. To ensure comprehension of task instructions, a practice run is undertaken before the actual test, which consists of a mixture of targets and distractor items displayed only along the midline of the page.To reliably identify forms of either egocentric, allocentric, or mixed forms of neglect, we referred to the normative values available on asymmetry scores. The asymmetry score for egocentric neglect corresponds to the difference between the number of correct targets cancelled in the left boxes (1 to 4) and the number of correct targets cancelled in the right boxes (7 to 10). Based on normative values [25], performances in which the difference between left and right total omissions is equal to or above a score of 3 is considered pathological. The asymmetry score for allocentric neglect corresponds to the difference between the total number of distractor apples cancelled with a left opening and the number of apples cancelled with a right opening. Based on normative values [25], a difference score equal to or above 2 was considered pathological. Individual performances in terms of asymmetry scores (above or below threshold) are presented in Table 1. Both pre- and post-PA training values are presented.For the purpose of carrying out quantitative analyses of pre- and post-PA training differences, we analyzed the number of omission or commission errors as a function of space, i.e., in boxes 1–2 (extreme left), 3–4, 5–6, 7–8, and 9–10 (extreme right).The *Behavioral Inattention Test* (BIT; [26]) is a battery of 15 tests designed to measure neglect: six “conventional” subtests and nine “behavioral” activities. In the present study, we only used the conventional subtests: Line Crossing, Letter Cancellation, Star Cancellation, Figure and Shape Copying, Line Bisection, and Representational Drawing. The stimuli and procedures of these individual subtests are provided in the BIT’s manual [26]. Wilson et al. (1987) also provide normative values for both individual subtests and the total score on conventional subtests. Individual performances scoring below the cut-off in the total score are presented in Table 1.To examine the effect of PA training, the total score on conventional subtests was also used as a dependent measure.The *Bells Test* [27] is a cancellation test in which 315 stimuli are pseudo-randomly scattered on an A4 sheet of paper. For the purpose of analysis, stimuli are divided in seven (ideal) columns: three on the left, one in the middle, and three on the right of the paper, each containing five bells. The patient has to identify and cancel out 35 bells from 280 distractors in a maximum of 5 min. To ensure comprehension of task instructions, a practice run is undertaken before the test administration; it includes a mixture of oversize targets and distractors. The participant is asked to name the elements in order to verify proper object recognition.The number of total omissions of targets and the difference between the omissions on the left three columns and the right three columns are scored. Mancuso et al. [28] provide normative values for both types of score. As for the total number of omissions, cut-off points considering the effect of age and education are provided (see Table 2 in Mancuso et al. [28]). As for the difference score, individual performances in which the difference between left and right total omissions is equal to or above 3 are considered pathological. Individual performances below these cut-offs are marked in Table 1.For the purpose of carrying out quantitative analyses of pre–post PA training differences, we analyzed the number of total omissions and the number of left omissions.

#### 2.2.2. Prismatic Adaptation Procedure

All participants were treated with the PA training, following the procedures described by Farnè et al. [23] and Frassinetti et al. [29]. It consists of 10 sessions, one per day, of about 30 min each, over a period of two weeks. Patients were seated in front of a table, with a wooden box (height 30 cm, central depth 34 cm and 18 cm at the periphery, width 72 cm) opened both on the experimenter’s and on the patient’s side. On the experimenter’s side, the box has a rounded edge, graduated in centimeters, in order to quantify the patient’s pointing accuracy. 

A visual target (e.g., a pen) was presented by the experimenter on the rounded edge of the box. Visual targets were presented 90 times, equally but randomly distributed in the middle, on the left and on the right side of the patient. Patients’ pointing spatial accuracy was recorded as the distance between the target position and the patient’s finger at the end of the pointing.

Patients were asked to keep their right (ipsilesional) hand on their chest, at the level of the sternum (hand starting position), and to point with the index finger toward the pen. The pointing arm movement was executed below the top face of the box, in order to hide the arm’s trajectory.

The pointing task was performed in three experimental conditions: pre-exposure, exposure, and postexposure.

Pre-exposure condition. In order to train the patient, the first session consisted of 30 pointing movements with a visible hand and 30 pointing movements with a hidden hand. From the second session onward, only 30 pointing movements with the hidden hand were given, with the aim of verifying the persistence of the PA effect from the previous session.

Exposure condition (prism adaptation). Patients were asked to rapidly point to the 90 targets presented in a quasi-random fixed sequence (30 targets at the center, 30 on the right, and 30 on the left), while wearing goggles with prismatic lenses that produce a shift of 10 degrees in the visual field toward the right side. During the exposure condition, the pointing movements were hidden by the top face of the box. Only the final part of the movement could be seen, when the index finger emerged beyond the far edge of the box. 

Post-exposure condition. Immediately after prism removal, patients performed 30 pointing movements with a hidden hand. The pointing movement was performed entirely below the top face of the box, so that the movement was not visible at any stage (invisible pointing).

### 2.3. Data Analyses

PA effects on neglect recovery were examined in two ways: by means of standard ANOVAs and by means of an alternative approach based on Mean Position of Hits calculation across physical space (MPH; [30]). In the former analysis, we entered the group factor, marking the three different forms of neglect (egocentric, allocentric, and mixed) as unrepeated factor and the effect of treatment (T_0_, T_1_) as repeated factor. Whenever appropriate, post hoc comparisons were carried out by means of the Bonferroni test.

Other factors could be included in the analysis, depending on the test considered. In particular, in the case of the Apples Cancellation Test, a repeated measure factor was space, with a five-level fractionation (boxes 1–2 on the extreme left, boxes 3–4 on left side, central boxes 5–6, box 7–8 on right side, and boxes 9–10 on the extreme right). One ANOVA was carried out on the total number of omissions as a dependent measure and a separate ANOVA on the total number of commission errors on the left-sided open apples. As the groups differed in the spatial distribution of both omission and commission errors (see below), we also carried out separate ANOVAs for the three groups of patients with different forms of neglect. Note that results for the groups of patients with egocentric or allocentric neglect should be seen as only exploratory due to the small size of the respective samples.

The analysis based on MPH allowed us to diagnose and quantify neglect by using the method proposed by Toraldo et al. [30]. This is based on the Mean Position of Hits (detected targets) calculation, an index that eliminates the confounding effects of non-lateral deficits that could decrease the hit rate. The procedure for MPH calculation offers the advantage of deriving a single, theoretically founded neglect index that captures the essence of a patient’s performance; in particular, this provided a measure of neglect severity which overcomes the conventional divisions of space (into halves of the display or into lateral columns or sectors, as done in the Bells and Apples tests). For these reasons, we used this index to analyze the differential effects of treatment on the severity of egocentric neglect vs. the severity of allocentric neglect. MPH was calculated from omissions/hit count in the Apples test in order to obtain the egocentric score, using the free software for MPH computation provided by Toraldo et al. [30]. We also extended the MPH calculation on commission errors; to this aim, the MPH values were computed by considering the space internal to the single apples. We attributed a −1 co-ordinate to every apple with a left-sided hole and a +1 co-ordinate to every apple with a right-sided hole, producing a reversed pass/fail score: a miss score (0) was given if an incomplete apple was marked; on the contrary, a hit score (1) was given if an incomplete apple was not marked. In order to test for differences in egocentric and allocentric neglect severity after PA treatment, we ran an ANOVA with treatment (T_0_, T_1_) and type of error (MPH obtained from omission/hit counts as an egocentric score, MPH obtained from commission/hit counts as an allocentric score) as repeated factors.

An ANOVA was carried out to examine performance on the BIT total conventional score as a dependent measure. As only one single measure was used, the analysis only considered the group and treatment factors.

Finally, two separate ANOVAs were carried out on the Bells test accuracy score (total omissions) and on the number of left omissions. Again, these analyses only considered the group and treatment factors.

## 3. Results

### 3.1. Neglect Recovery Following PA training: The Apples Cancellation Test

Figure 1 shows omission and commission errors before and after the PA training as a function of space separately for patients with different types of neglect (egocentric, allocentric, or mixed). Inspection of the figure indicates a few main findings:In patients with egocentric neglect, errors are confined to omissions and tend to increase going from right to left portions of space. The effect of PA is clear and tends to reduce (although not abolish) the left-right asymmetry;In patients with allocentric neglect, errors are predominantly commissions. No clear effect of space is apparent, as errors tend to be similar across all tested positions. No clear effect of training is apparent;In patients with mixed neglect, the pattern for omission errors is similar to that of patients with egocentric neglect due to the effect of the PA training. Commission errors tend to be more frequent in the right part of space; they also tend to increase slightly after the PA training.

In the analysis of omission errors, the ANOVA showed a main effect of treatment (F _(1,25)_ = 12.7; *p* = 0.001; η^2^ = 0.33): the number of targets omitted decreased from 4.6 (SE = 0.57) before treatment to 3.1 (SE = 0.60) after treatment. The main effect of group was not significant (F _(2,25)_ = 1.33; *p* = 0.28; η^2^ = 0.09). The significant main effect of space (F _(4,100)_ = 25.2; *p* = 0.000; η^2^ = 0.5) indicated a decreasing number of complete apples omitted proceeding from the extreme left boxes (mean = 5.7; SE = 0.66) to the extreme right ones (mean = 1.8; SE = 0.43). The space x group interaction (F _(8,100)_ = 4.8; *p* = 0.000; η^2^ = 0.27) showed that groups differed as a function of space: patients with allocentric neglect showed fewer omissions than patients with mixed neglect in boxes 1–2 (mean = 3.2, SE = 1.3 vs. mean = 7.4, SE = 0.74) and in boxes 3–4 (mean = 2.9, SE = 1.3 vs. mean = 6.5, SE = 0.7). Patients with egocentric neglect did not differ significantly from the two other groups, although, in the extreme left boxes, they showed a trend for more omissions with respect to patients with allocentric neglect (mean = 6.5, SE = 1.2 vs. mean = 3.2, SE = 1.3; *p* = 0.08). The other interactions were not significant.

In the ANOVA, on commission errors, there was no main effect of treatment (F _(1,25)_ = 0.69; *p* = 0.41; η^2^ = 0.02). The significant main effect of space (F _(4,100)_ = 3.8; *p* = 0.006; η^2^ = 0.13) indicated an increasing number of commission errors proceeding from the extreme left boxes (mean = 2.1, SE = 0.43) to the extreme right ones (mean = 4.01, SE = 0.78). The main effect of group (F _(2,25)_ = 10.09; *p* = 0.001; η^2^ = 0.44) indicated that patients with allocentric neglect made more commission errors with respect to patients with egocentric or mixed forms of neglect. 

Below, we present similar ANOVAs, separately carried out in the three groups of patients, with different forms of neglect. 

#### 3.1.1. Analysis on Patients with Egocentric Neglect

In the subsample of patients with egocentric neglect (*n* = 6), only the analysis on omission errors was carried out. The main effect of treatment was significant (F _(1,5)_ = 6.6; *p* = 0.05; η^2^ = 0.56), indicating that the number of complete apples omitted tended to decrease from 5.3 (SE = 0.9) before treatment to 3.1 (SE = 1.3) after treatment. The main effect of space was significant (F _(4,20)_ = 9.8; *p* = 0.000; η^2^ = 0.66), indicating an increasing number of omissions proceeding from the extreme right boxes (mean boxes 9–10 = 2.08, SE = 1.1) to the extreme left ones (mean boxes 1–2 = 6.5, SE = 1.3). No significant treatment x space interaction was detected (F _(4,20)_ = 1.2; *p* = 0.34; η^2^ = 0.19).

At an individual level, it may be noted that three out of six patients did not show a pathological asymmetry score for egocentric neglect after PA treatment (see Table 1).

#### 3.1.2. Analysis on Patients with Allocentric Neglect

In the subsample of patients with allocentric neglect (*n* = 5), only the analysis on commission errors was carried out. There were no significant main effects of treatment (F _(1,4)_ = 0.04; *p* = 0.84; η^2^ = 0.01) and space (F _(4,16)_ = 0.35; *p* = 0.83; η^2^ = 0.82). The treatment x space interaction was not significant (F _(4,16)_ = 0.27; *p* = 0.89; η^2^ = 0.06).

At an individual level, all five patients still showed a pathological asymmetry score for allocentric neglect after PA treatment (see Table 1).

#### 3.1.3. Analysis on Patients with Mixed Form of Neglect 

In the subsample of patients with mixed neglect (*n* = 17), the ANOVA on omission errors showed the main effect of treatment (F _(1,16)_ = 18.8; *p* = 0.001; η^2^ = 0.54), indicating that errors decreased from 5.8 (SE = 0.53) before treatment to 3.6 (SE = 0.56) after treatment. The main effect of space (F _(4,64)_ = 46.1; *p* = 0.000; η^2^ = 0.74) indicated an increasing number of omissions proceeding from the extreme right boxes (mean boxes 9–10 = 1.23, SE = 0.28) to the extreme left ones (mean boxes 1–2 = 7.4, SE = 0.63). The treatment x space interaction was significant (F _(4,64)_ = 3.18; *p* = 0.01; η^2^ = 0.16): Bonferroni post hoc comparisons revealed progressively fewer omissions in the post-treatment with respect to the pretreatment in all boxes (boxes 1–2: mean T_0_ = 8.7, SE = 0.57 – mean T_1_= 6.1, SE = 1; boxes 3–4: mean T_0_ = 8,SE = 0.74 – mean T_1_= 5.05, SE = 0.83; boxes 5–6: mean T_0_ = 7, SE = 0.86 – mean T_1_= 3.4, SE = 0.78; boxes 7–8: mean T_0_ = 4.3, SE = 0.81 – mean T_1_ = 2.2, SE = 0.56) except for those on the extreme right (boxes 9–10; *p* > 0.05).

At an individual level, it may be noted that three out of seventeen patients did not show a pathological asymmetry score for egocentric neglect after the PA treatment (see Table 1).

The ANOVA on commission errors showed the main effect of space (F _(4,64)_ = 9.7; *p* = 0.000; η^2^ = 0.37), indicating a tendency for errors to be more frequent in the right part of space: patients made fewer commission errors in the extreme left boxes (mean boxes 1–2 = 1.05; SE = 0.41) with respect to boxes 5–6 (mean = 2.02, SE = 0.47), boxes 7–8 (mean = 3.2, SE = 0.67) and 9–10 (mean = 5.1; SE = 1.06). No main effect of treatment was present (F _(1,16)_ = 0.81; *p* = 0.37; η^2^ = 0.49). The treatment x space interaction was not significant (F _(4,64)_ = 0.6; *p* = 0.66; η^2^ = 0.03). 

### 3.2. The Effect of Treatment on Mean Position of Hits (MPH) Index 

In the ANOVA on egocentric and allocentric severity score, there was no main effect of treatment (F _(1,27)_ = 3.2; *p* = 0.08; η^2^ = 0.1). The main effect of type of error was not significant (F _(1,27)_ = 3.2; *p* = 0.08; η^2^ = 0.1). The treatment x type of error interaction was significant (F _(1,27)_ = 10.3; *p* = 0.003; η^2^ = 0.27). The asymmetry for omission errors significantly decreased from baseline to post-treatment (mean score T_0_ = 0.18, SE = 0.02 vs. mean score T_1_ = 0.10, SE = 0.02; *p* = 0.004), while asymmetry for commission errors did not differ from baseline to post-treatment (mean score T_0_ = 0.06, SE = 0.02 vs. mean score T_1_ = 0.09, SE = 0.02; p > 0.05).

### 3.3. Effect of PA in the Other Neglect Tests

In the ANOVA on the BIT (conventional) total score, there was a main effect of treatment (F _(1,25)_ = 7.8; *p* = 0.01; η^2^ = 0.23): the BIT score was significantly higher in the post-treatment (mean score T_1_ = 120.1, SE = 5.5) than at the baseline (mean score T_0_ = 103.1, SE = 6.7). Neither the main effect of the group factor (F _(1,25)_ = 1.3; *p* = 0.27; η^2^ = 0.09) nor the group by treatment interaction (F _(2,25)_ = 0.09; *p* = 0.91; η^2^ = 0.007) were significant.

Separate ANOVAs on the Bells test were carried out on the accuracy score (total omissions) and on the number of left omissions. There was a significant main effect of treatment on total omissions (F _(1,25)_ = 9.98; *p* = 0.004; η^2^ = 0.28), as well as on left-sided omissions (F _(1,24)_ = 10.3; *p* = 0.004; η^2^ = 0.3). Total omissions (mean score T_0_ = 14.8, SE = 1.8) appreciably decreased after prismatic training (mean score T_1_ = 8.1; SE = 1.8); similarly, left omissions reduced from pre-training (mean T_0_ = 9.2, SE = 1.08) to post-training (T_1_ mean = 5.2, SE = 1.2). No significant effect of group was found on total accuracy score (total omissions) (F _(2,25)_ = 1.9; *p* = 0.16; η^2^ = 0.13) and on left omissions (F _(2,24)_ = 1.6; *p* = 0.22; η^2^ = 0.11). The group by treatment interaction was not significant in either analysis (total omissions: F _(2,25)_ = 0.33; *p* = 0.72; η^2^ = 0.02; left omissions: F _(2,24)_ = 0.25; *p* = 0.77; η^2^ = 0.02).

## 4. Discussion

The results generally confirmed the effectiveness of the PA training in ameliorating neglect symptoms. After the intervention, as a whole group, patients were generally better at performing the conventional tasks of the BIT, cancelling out more bells and complete apples in the respective tests. By contrast, no change was detected for the number of commission errors in the Apples Cancellation test following PA treatment. These results are in keeping with the idea that PA training is effective in ameliorating egocentric, but not allocentric, neglect [13,14].

Tests such as the Bells and the Apples tests envisage a conventional division of space into evenly distributed sectors or columns. While this approach is effective for descriptive purposes, it fails to provide a single measure of neglect severity. To this aim, we used an index (MPH) which was developed to overcome this limitation [30]. This allowed us to obtain a single measure of egocentric, as well as allocentric, neglect severity with comparable metrics. The data obtained from the MPH index calculation from omission/hit counts confirmed the selective effect of PA treatment on the egocentric component of neglect. By contrast, the MPH computation on commission errors indicated no effect of PA treatment on allocentric neglect severity. Overall, the computation based on the MPH indexes [30] closely confirmed the dissociation between changes in egocentric versus allocentric severity of neglect as a function of PA training. This finding indicates that the dissociation observed was not due to the reference to a conventional categorical subdivision of space, as adopted in the Apples test [15].

As expected, the effect of the rehabilitative treatment varied as a function of the type of neglect. Patients with egocentric neglect or with mixed neglect showed significant improvements in performance of the Apples Cancellation test, while patients with allocentric neglect did not show any detectable change after treatment. Thus, separating patients according to their form of neglect is useful in understanding their performance as a function of PA treatment.

Reduction of omission errors with extension of exploration toward the left is in accordance with the documented improvement, following PA, of neglect around the center of reference system for spatial coding [31]. Prism adaptation may shift the egocentric coordinates of a sensory–motor reference frame (realignment), bringing at least part of the neglected hemispace into the dysfunctional task-workspace. In this way, PA partially compensates for the dysfunctional positioning without acting on the sizing of a task-workspace. By contrast, the present findings indicate that the PA training is not effective in reducing the allocentric component of neglect, thus confirming previous evidence [13,14]. 

To separate patients according to their form of neglect, we referred to the Apples Cancellation test [15]. Based on the performance in this test, we could identify patients who only made omissions on the left side of space (egocentric neglect), as well as patients who only made commission errors failing to detect the missing part of the apples on the left side (allocentric neglect), regardless of the location of the target on the paper sheet. Both groups were comparatively small, accounting for 21.4% and 17.8% of the examined patients, respectively. Many more patients (60.7%) showed a mixed pattern, characterized by the co-presence of omission and commission errors. Clearly, the size of the present sample is too small to definitely establish proportions, but the figures obtained here are comparatively similar to other data in the literature. For example, out of a group of 59 patients with right-sided damage, 32 showed some signs of asymmetric exploration [15]. Out of these, six patients showed isolated egocentric neglect for left space (18.7%), six isolated allocentric neglect on the left, and 15 patients (46.9%) showed both asymmetries (smaller proportions of patients showed neglect for right parts of space or objects). 

In general, these data call for at least two main observations. On the one hand, patients with a pure form of allocentric neglect are comparatively few. On the other, however, most patients with seemingly egocentric neglect also appear to have allocentric difficulties, a finding which is presumably overlooked in the interpretation of the disorder. Indeed, it seems likely that allocentric difficulties may go undetected unless dedicated instruments (such as the Apples Cancellation test) are used. This may actually be clinically relevant, as it has been reported that the presence of allocentric neglect was associated with lower scores on the Barthel Index and greater difficulties in activities of daily life [15].

In this vein, it is of note that the presence of difficulties in the allocentric frame of reference may interact in complex ways in the performance of behavioral tasks commonly used to diagnose neglect. First of all, the interaction between body- and object-centered reference frames is clear in the case of the performance in the Apples Cancellation test itself. Thus, while patients with allocentric neglect showed commission errors that were independent of space, those with mixed neglect showed a decreasing number of commission errors going from right to left space. Presumably, this indicates that reduced exploration of left space leaves less room for the possibility of making commission errors.

However, this interaction may also influence performance in other neglect tests. For example, the BIT battery includes conventional tasks such as Figure and Shape Copying, Line Bisection, and Representational Drawing, in which a failure in object-centered coordinates can appreciably affect performance [26]. Indeed, out of the five patients with allocentric neglect, all but one failed in the BIT total conventional score. Somewhat more difficult is to adjudicate the role of allocentric difficulties in cancellation tasks, such as the Bells test or the Line Crossing, Letter Cancellation, and Star Cancellation subtests of the BIT, in which a scattered presence of pathological performance was detected. One possibility is that the inability to fully appreciate each individual target (among the many typically displayed in cancellation tasks) can foster the emergence of exploration asymmetries even in the case of subclinical egocentric disorders, particularly in the case of more complex target-distractor discrimination. However, an alternative hypothesis is that tasks such as the Bells test or the Star Cancellation subtest are simply more sensitive in detecting mild egocentric deficits present in these patients. The availability of more instruments focusing on the object-centered reference frame would certainly be important in more clearly defining the allocentric disturbance, which is still poorly understood [16].

Some limits of the present study should be addressed. Even though we tried to recruit a comparatively large sample of patients with neglect, the number of patients in the critical egocentric and allocentric subgroups was certainly too small to draw definitive conclusions. Thus, further research on the effect of PA treatment on patients with pure egocentric or allocentric neglect is certainly in order to confirm the present results. Furthermore, this would allow for a finer-grain evaluation of the impact of PA treatment. For example, in the present study, the six patients with egocentric neglect showed a particularly large improvement in performance after training (η^2^ = 0.56), and three out of six actually scored within normal limits at the post-test evaluation. The corresponding improvements in patients with mixed neglect was similar (η^2^ = 0.54); however, a smaller proportion of patients (3 out of 17) performed within normal limits at the post-test evaluation. This pattern raises the interesting possibility that the presence of allocentric neglect actually limits the effectiveness of the PA treatment over the egocentric component itself. Clearly, the present data are insufficient to draw such a conclusion, but this data-driven hypothesis may deserve to be checked with a larger sample of patients with neglect.

Another limit of the present study concerns the evaluation of the object-centered reference frames. Only a few instruments allow for the detection of allocentric forms of neglect. Here, we relied on the Apples Cancellation Test, which has the advantage of norms based on a relatively large sample of control subjects [25]. However, relying only on such a test, we were forced to use it both for classifying the different forms of neglect and for measuring the quantitative changes as an effect of PA treatment. This approach is certainly not optimal, and research using a greater variety of instruments assessing object-centered neglect would be certainly important to confirm the dissociation between egocentric and allocentric modifications following PA treatment. To partially compensate for this problem, we also used an index which provides a single measure of egocentric and allocentric neglect severity independently from the conventional divisions of space into sectors or columns. The results based on the MPH index provided support for the dissociation between egocentric and allocentric forms of neglect.

## 5. Conclusions

Our data indicate that the PA treatment selectively improves egocentric neglect, whereas it is ineffective in modifying the allocentric component of the disorder. Notably, the allocentric component of neglect is frequently impaired, although most often in conjunction with the egocentric one, yielding the mixed form of neglect detected in most patients. This finding stresses the importance of developing exercises tuned to improve the “neglected” allocentric component of neglect.

## Figures and Tables

**Figure 1 brainsci-09-00327-f001:**
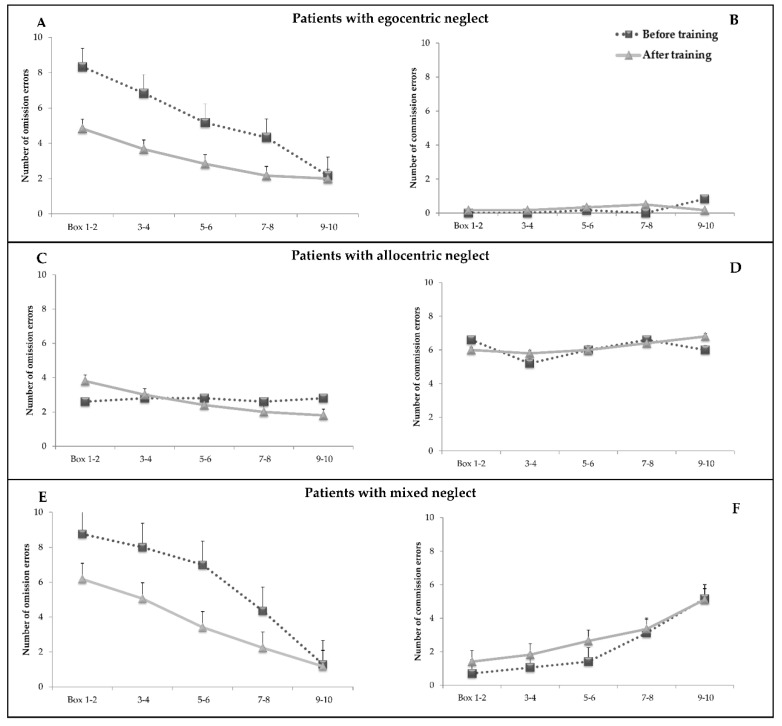
Mean number of omission (left) and commission (right) errors before and after the PA training as a function of space (bars indicate standard errors). Boxes 1–2 are on the extreme left, then boxes 3–4, and so on. Plot (**A**) shows the number of complete apples omitted (omission errors) as a function of space in the group of patients with egocentric neglect (*n* = 6). Plot (**B**) shows the number of apples with left-sided holes erroneously marked as a function of space (commission errors). Plots (**C**) and (**D**) show the same data for the group of patients with allocentric neglect (*n* = 5). Plots (**E**) and (**F**) show the same data for the group of patients with mixed neglect (*n* = 17).

**Table 1 brainsci-09-00327-t001:** Individual performance in the Behavioural Inattention Test (BIT) conventional, in the Bells test, and in the Apples Cancellation test before (T_0_) and after (T_1_) the PA training. A + sign indicates a pathological performance according to the norms (in the case of BIT, a + indicates a low performance in at least two subtests of the scale; a – sign indicates a performance within normal limits).

	T0					T1				
	BIT Conventional	Bells Test		Apples Cancellation Test		BIT Conventional	Bells Test		Apples Cancellation Test	
Patient ID	Subtest Score	Total Accuracy Score	Asymmetry Score	Asymmetry Score for Egocentric Neglect	Asymmetry Score for Allocentric Neglect	Total Score	Total Accuracy Score	Asymmetry Score	Asymmetry Score for Egocentric Neglect	Asymmetry Score for Allocentric Neglect
**Patient with egocentric neglect**										
**Pt 1**	+	+	-	+	-	-	-	-	-	-
**Pt 2**	+	+	+	+	-	-	-	-	-	-
**Pt 3**	+	+	+	+	-	+	-	-	+	-
**Pt 4**	+	+	+	+	-	-	+	-	-	+
**Pt 5**	-	+	+	+	-	+	+	+	+	+
**Pt 6**	+	+	+	+	-	+	+	+	+	-
**Patients with allocentric neglect**										
**Pt 1**	+	-	-	-	+	+	+	-	-	+
**Pt 2**	+	+	+	-	+	-	-	+	+	+
**Pt 3**	+	+	+	-	+	-	+	+	-	+
**Pt 4**	+	+	-	-	+	+	+	-	+	+
**Pt 5**	+	-	-	-	+	-	-	-	+	+
**Patients with mixed neglect**										
**Pt 1**	+	+	+	+	+	+	+	+	+	+
**Pt 2**	+	+	+	+	+	+	+	+	+	+
**Pt 3**	+	+	+	+	+	+	+	+	+	+
**Pt 4**	+	+	+	+	+	-	-	-	-	+
**Pt 5**	+	+	+	+	+	-	-	-	+	-
**Pt 6**	+	+	+	+	+	+	+	+	+	+
**Pt 7**	+	+	+	+	+	-	-	-	+	+
**Pt 8**	+	+	+	+	+	+	+	+	+	+
**Pt 9**	+	+	+	+	+	+	+	+	+	+
**Pt 10**	+	+	-	+	+	+	+	-	-	-
**Pt 11**	+	+	+	+	+	+	+	+	+	+
**Pt 12**	+	+	+	+	+	+	-	-	+	+
**Pt 13**	+	+	+	+	+	-	-	-	-	-
**Pt 14**	+	+	+	+	+	-	-	+	+	-
**Pt 15**	+	+	+	+	+	+	+	+	+	+
**Pt 16**	+	+	-	+	+	+	+	-	+	+
**Pt 17**	+	+	+	+	+	+	+	+	+	-

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
