# Peer review of "Effects of Prism Adaptation on Reference Systems for Extrapersonal Space in Neglect Patients"

_brainsci, 2019, doi:10.3390/brainsci9110327_

Round 1

Reviewer 1 Report

The authors successfully addressed all of my points, so I am happy to accept the paper

Some minor checking on the language might be in order, e.g., line 226, "ran" instead of "run"

Author Response

Point 1: The authors successfully addressed all of my points, so I am happy to accept the paper.

Response 1: We thank the reviewer for accepting the manuscript. We are very happy to have successfully addressed all your points.

Point 2: Some minor checking on the language might be in order, e.g., line 226, "ran" instead of "run"

Response 2: We modified "run" in line 226 in "ran".

Reviewer 2 Report

The authors fully addressed all the concerns previously raised.

Author Response

Point 1: The authors fully addressed all the concerns previously raised.

Response 1: We thank the reviewer. We are happy to have fully addressed all the concerns raised.

This manuscript is a resubmission of an earlier submission. The following is a list of the peer review reports and author responses from that submission.

Round 1

Reviewer 1 Report

Thanks for an interesting piece of evidence. The authors report a study that investigates whether prism adaptation can be useful to rehabilitate egocentric vs allocentric neglect. While perhaps not really new – after all, prism adaptation is a manoeuvre that acts on egocentric visuo-motor mechanisms, so the absence of allocentric effects is expected and to my knowledge has already been reported, especially concerning allocentric tasks line line bisection – replication and consolidation of results is always welcome, especially in the current period in which, at last, the value of replication/consolidation begins to be appreciated in the neuropsychological community. In this vein, I believe this work should finally warrant publication, albeit the authors should improve it in some critical parts. I list my main remarks below.

Exclusion of patients with hemianopia

Many patients with neglect also have hemianopia; given that the purpose of this work is to provide suggestions/insight as to the rehabilitation of neglect, I am not sure what the rationale is for this exclusion. The authors should justify their choice.

Measurement and statistics

In short, I do think the results are interesting, but I am not sure the current analyses give a transparent picture of them. Some choices are suboptimal, so the question that arises is whether results would look the same after some more straightforward analyses have been carried out.

The authors are interested in the impact that treatment should have on, essentially, two complementary measures, one of egocentric neglect and another of allocentric neglect. The most straightforward way to answer this question would have been to analyse the differential effects of treatment on the severity of egocentric neglect vs the severity of allocentric neglect. Thus, one would need to choose a single measure of egocentric neglect severity, a single measure of allocentric neglect severity (with comparable metric), and show that there is a significant PRE-POST by MEASURE (ego- vs allo-) interaction.

By contrast, the authors divided patients in three subgroups on ground of the performance they showed in the pre-treatment phase. This is a non-standard procedure –indeed in general it is not advisable to group subjects on grounds of the very same variables one is willing to analyse as dependent, as this is likely to produce loopholes with unpredictable or complex drawbacks (which only extensive Monte Carlo simulation studies would uncover). Unless strictly necessary, I would rely on the simpler, more transparent design I detailed above. At first glance, I do not see obvious reasons why one should group patients – if anything, I see reasons not to group them at all. In the current climate in which (at last!) power limitations have been recognized as responsible for the serious inflation of false discovery rates in (neuro)psychology, losing power with unnecessary grouping should be avoided. Statistical conclusions based on groups of 5 or 6 individuals are very seriously at risk of generating false positives. But this is not really a problem with the present study, because the authors collected a reasonable number (28) of patients, so, why grouping them? They can, and I believe they should, try to show a change of the egocentric neglect score with treatment (or the absence thereof for the allocentric measure) on the N=28 sample taken as a whole. Perhaps I am missing something here, but if so, the authors should fully report strong theoretical justification for running an analysis which mixes up independent and dependent variables and wastes so much power.

Another complication which can be avoided is dividing space in five sectors and analysing it as a categorical variable in ANOVA. There are two problems with this choice. First, space is a continuous variable and treating it as categorical will fatally increase the number of  comparisons (again the problem of power loss and inflation of false discovery rate). One possible suggestion would be that of running within-patient regressions across space, but that would produce very instable parameters; moreover, ANOVA assumes homoscedasticity and Gaussian shape of residuals, both of which are undoubtedly false given that the dependent variable is error count, with clear-cut ceiling and floor effects. Instead of splitting performance in five sectors, the authors can use single, summary measures which represent neglect severity in an unambiguous and theoretically sound way. The authors refer to neglect work which uses, essentially, differences between left and right counts of errors. This measure has been shown to be flawed or distorted – see e.g. Rorden and Karnath, Neuropsychologia 2010; Toraldo et al., Clinical Neuropsychologist 2017; Huygelier & Gillebert, J of Neuropsychology 2018). By contrast, the Mean Positon of Hits (MPH) is a measure with optimal psychometric properties and would nicely serve the purpose of the authors – it is very far from its upper and lower bounds and has been shown to follow an approximately normal curve (justifying GLM or ANOVA and the like). A more transparent and data-dredging-free way to make the authors’ point would be that of running a GLM with pre-post and egocentric/allocentric as between subjects predictors, and with MPH as dependent variable. Indeed MPH can be obtained both from omission/hit counts, leading to the egocentric score (see Toraldo et al., 2017), and from commission errors, leading to the allocentric score. In the latter case, my best guess is that one should consider the space internal to the single apple, giving a left-sided hole a -1 co-ordinate and a right-sided hole a +1 co-ordinate, and giving reversed pass/fail scores: if the patient marks an incomplete apple, the inference is that the hole was “neglected”, so a 0 (miss) score should be given in this case, and 1 (hit) score should be given to unmarked incomplete apples.

As for the allocentric measure that is reported in Fig. 1 – number of commission errors, I read in the Figure legend that the authors only counted the cases of crossed-out apples with left-sided holes. Commission errors can reflect not only allocentric neglect, but any sort of other causes, e.g. amblyopia, visual form agnosia, dis-executive syndromes, perseveration, or, as the authors themselves correctly argued, the mere availability of space to explore. These factors would all produce a generic increase in commission errors of both kinds, i.e. both left-holed and right-holed apples – and this would explain the strange increase of commission errors on the right side of the display in terms of availability of space that the patient can effectively visit (bottom-right panel of Fig. 1). To show whether or not this criticism applies, the authors should report and analyse the difference between right-hole and left-hole commission errors rather than the simple count of the former. Of course, separate MPHs for the five columns can be derived to make this analysis consistent with the one suggested earlier.

Minor points

Fig. 1 Draw standard errors from the two plots (before and after training) in such a way that they do not overlap.

Author Response

Point 1: Exclusion of patients with hemianopia

Many patients with neglect also have hemianopia; given that the purpose of this work is to provide suggestions/insight as to the rehabilitation of neglect, I am not sure what the rationale is for this exclusion. The authors should justify their choice.

Response 1: Thank you. We added in the manuscript the rationale for exclusion of patients with hemianopia.  As reported in the work of Serino et al. (2006), the process of neglect improvement induced by PA, probably mediated by the reorganization of the oculo-motor system, needs to be guided by visual information. The study of patients’ neuroanatomical data showed that severe occipital lesions were associated with a lack of error reduction, poor neglect recovery and reduced oculo-motor system amelioration. So in the case of patients with hemianopia, the visual field deficit didn’t allow the documented low-order visuo-motor reorganization induced by PA that promotes a resetting of the oculo-motor system leading to an improvement in high-order visuo-spatial representation able to ameliorate neglect.

Point 2: Measurement and statistics

In short, I do think the results are interesting, but I am not sure the current analyses give a transparent picture of them. Some choices are suboptimal, so the question that arises is whether results would look the same after some more straightforward analyses have been carried out.

The authors are interested in the impact that treatment should have on, essentially, two complementary measures, one of egocentric neglect and another of allocentric neglect. The most straightforward way to answer this question would have been to analyse the differential effects of treatment on the severity of egocentric neglect vs the severity of allocentric neglect. Thus, one would need to choose a single measure of egocentric neglect severity, a single measure of allocentric neglect severity (with comparable metric), and show that there is a significant PRE-POST by MEASURE (ego- vs allo-) interaction.

By contrast, the authors divided patients in three subgroups on ground of the performance they showed in the pre-treatment phase. This is a non-standard procedure –indeed in general it is not advisable to group subjects on grounds of the very same variables one is willing to analyse as dependent, as this is likely to produce loopholes with unpredictable or complex drawbacks (which only extensive Monte Carlo simulation studies would uncover). Unless strictly necessary, I would rely on the simpler, more transparent design I detailed above. At first glance, I do not see obvious reasons why one should group patients – if anything, I see reasons not to group them at all. In the current climate in which (at last!) power limitations have been recognized as responsible for the serious inflation of false discovery rates in (neuro)psychology, losing power with unnecessary grouping should be avoided. Statistical conclusions based on groups of 5 or 6 individuals are very seriously at risk of generating false positives. But this is not really a problem with the present study, because the authors collected a reasonable number (28) of patients, so, why grouping them? They can, and I believe they should, try to show a change of the egocentric neglect score with treatment (or the absence thereof for the allocentric measure) on the N=28 sample taken as a whole. Perhaps I am missing something here, but if so, the authors should fully report strong theoretical justification for running an analysis which mixes up independent and dependent variables and wastes so much power.

Another complication which can be avoided is dividing space in five sectors and analysing it as a categorical variable in ANOVA. There are two problems with this choice. First, space is a continuous variable and treating it as categorical will fatally increase the number of comparisons (again the problem of power loss and inflation of false discovery rate). One possible suggestion would be that of running within-patient regressions across space, but that would produce very instable parameters; moreover, ANOVA assumes homoscedasticity and Gaussian shape of residuals, both of which are undoubtedly false given that the dependent variable is error count, with clear-cut ceiling and floor effects. Instead of splitting performance in five sectors, the authors can use single, summary measures which represent neglect severity in an unambiguous and theoretically sound way. The authors refer to neglect work which uses, essentially, differences between left and right counts of errors. This measure has been shown to be flawed or distorted – see e.g. Rorden and Karnath, Neuropsychologia 2010; Toraldo et al., Clinical Neuropsychologist 2017; Huygelier & Gillebert, J of Neuropsychology 2018). By contrast, the Mean Positon of Hits (MPH) is a measure with optimal psychometric properties and would nicely serve the purpose of the authors – it is very far from its upper and lower bounds and has been shown to follow an approximately normal curve (justifying GLM or ANOVA and the like). A more transparent and data-dredging-free way to make the authors’ point would be that of running a GLM with pre-post and egocentric/allocentric as between subjects predictors, and with MPH as dependent variable. Indeed MPH can be obtained both from omission/hit counts, leading to the egocentric score (see Toraldo et al., 2017), and from commission errors, leading to the allocentric score. In the latter case, my best guess is that one should consider the space internal to the single apple, giving a left-sided hole a -1 co-ordinate and a right-sided hole a +1 co-ordinate, and giving reversed pass/fail scores: if the patient marks an incomplete apple, the inference is that the hole was “neglected”, so a 0 (miss) score should be given in this case, and 1 (hit) score should be given to unmarked incomplete apples.

As for the allocentric measure that is reported in Fig. 1 – number of commission errors, I read in the Figure legend that the authors only counted the cases of crossed-out apples with left-sided holes. Commission errors can reflect not only allocentric neglect, but any sort of other causes, e.g. amblyopia, visual form agnosia, dis-executive syndromes, perseveration, or, as the authors themselves correctly argued, the mere availability of space to explore. These factors would all produce a generic increase in commission errors of both kinds, i.e. both left-holed and right-holed apples – and this would explain the strange increase of commission errors on the right side of the display in terms of availability of space that the patient can effectively visit (bottom-right panel of Fig. 1). To show whether or not this criticism applies, the authors should report and analyse the difference between right-hole and left-hole commission errors rather than the simple count of the former. Of course, separate MPHs for the five columns can be derived to make this analysis consistent with the one suggested earlier.

Response 2: We thank the reviewer for the useful suggestions. We tried to diagnose and quantify neglect using the method proposed by Toraldo et al. (2017). We analysed the differential effects of treatment on the severity of egocentric neglect vs the severity of allocentric neglect. MPH was calculated from omissions/hit count in the Apple test in order to obtain the egocentric score. We also tried to extend MPH calculation on commission errors, so the MPHs were computed considering the space internal to the single apple. As suggested, we attributed a - 1 co-ordinate to a left-sided hole and a + 1 co-ordinate to a right-sided hole, producing a reversed pass/fail scores: a miss score (0) was given if an incomplete apple was marked, a hit score (1), on the contrary, was given if an incomplete apple was not marked. In order to test differences in egocentric and allocentric neglect severity after PA treatment, we run an ANOVA with treatment (T0, T1) and type of error (MPH obtained from omission/hit counts for egocentric score, MPH obtained from commission/hit counts for allocentric score) as repeated factors.

In the ANOVA on egocentric and allocentric severity score there was no main effect of treatment (F (1,27) = 3.2; p = 0.08; η2 = 0.1). The main effect of type of error was not significant (F (1,27) = 3.2; p = 0.08; η2 = 0.1). The treatment x type of error interaction was significant (F (1,27) = 10.3; p = 0.003; η2 = 0.27). The asymmetry for omission errors significantly decreased from baseline to post-treatment (Mean score T0 = 0.18, SE = 0.02 vs Mean score T1 = 0.10, SE = 0.02; p = 0.004), while asymmetry for commission errors did not differ from baseline to post-treatment (Mean score T0 = 0.06, SE = 0.02 vs Mean score T1 = 0.09, SE = 0.02; p > .05).

We have added these data to the Results section of the paper. Furthermore, we added a paragraph in the discussion section to illustrate these new findings.

Point 3: Minor points

Fig. 1 Draw standard errors from the two plots (before and after training) in such a way that they do not overlap.

Response 3: Thank you. We draw standard errors in the plots.

Reviewer 2 Report

This study presents interesting neuropsychological data on the effects of PA training on subjects with egocentric, allocentric and mixed neglect. In short, PA training improves the performance of patients with egocentric but not allocentric neglect. The work is well written. I have only one major point, I hope the authors may clarify.

In my opinion, it would be necessary to argue/explain better the research question behind this study. In other words, the authors need to explain why they expect PA training to have an effect on patients with allocentric neglect. In fact, as the authors themselves admit in their discussions (page 5 "Prism adaptation may shift the egocentric coordinates of a sensory-motor reference frame (realignment), bringing at least part of the neglected hemispace into the dysfunctional task-work space. ") the PA training would play a role on egocentric components. So why expect effects on the allocentric component? Probably this is due to the fact that the authors in the introduction state "The authors attributed this phenomenon to damage of the dorsal way and to lesions in the post-central cortical areas correlated with the allocentric frame of reference". However, the literature on reference systems reports that the occipito-temporal areas support allocentric representations (e.g. Ruotolo et al., 2019; Galati et al., 2000; Committeri et al., 2004; Milner & Goodale, 1995, 2008). 

Finally, another minor point concerns the abstract. Authors state: "However, the allocentric component of neglect is frequently impaired, although this is most often in conjunction with the egocentric impairment". What exactly do the authors mean? The fact that the allocentric component is damaged and does not improve after training was already clear before.

Author Response

Point 1: In my opinion, it would be necessary to argue/explain better the research question behind this study. In other words, the authors need to explain why they expect PA training to have an effect on patients with allocentric neglect. In fact, as the authors themselves admit in their discussions (page 5 "Prism adaptation may shift the egocentric coordinates of a sensory-motor reference frame (realignment), bringing at least part of the neglected hemispace into the dysfunctional task-work space. ") the PA training would play a role on egocentric components. So why expect effects on the allocentric component? Probably this is due to the fact that the authors in the introduction state "The authors attributed this phenomenon to damage of the dorsal way and to lesions in the post-central cortical areas correlated with the allocentric frame of reference". However, the literature on reference systems reports that the occipito-temporal areas support allocentric representations (e.g. Ruotolo et al., 2019; Galati et al., 2000; Committeri et al., 2004; Milner & Goodale, 1995, 2008). 

Response 1: Thank you for this suggestion. As stated in the introduction, current studies indicate an effect of PA training on egocentric neglect but less or no effect on allocentric neglect. Since evidence about PA effects on allocentric neglect is still limited to the only study that investigated the effect of PA treatment on post-acute patients, using only four session of PA, we aimed to clarify PA effects on patients with different reference frame deficit. To our knowledge, Goosmann’s et al. study (2008) was the first that detected an increasing allocentric neglect in post-acute patients after a short period of PA training. The training showed a positive effect on ego-centric aspects of neglect, with no significant effects on allocentric ones. Some previous studies (Ferber et al., 2003; Striemer & Danckert, 2010) yielded similar results in compar­ing the short-term experimental effects of PA on chimerical faces and objects, arguing that performance in such tasks cannot be improved by wearing prism glasses. The dissociation between perceptual and motor influences of PA was highlighted by the shifting induced by the treatment in eye movements without effects on the perception of stimuli in neglected space. These data are consistent with the notion that PA influences dorsal stream processes without influencing perceptual biases in the ventral stream in patients with neglect (Striemer & Danckert, 2010), so that while the exploratory eye movements of a patient with neglect were clearly shifted toward the left after PA, he still showed no awareness for the left side of the stimuli he was now actively exploring (Ferber et al., 2003). However, all these studies involved only patients with mixed forms of neglect, without a distinction according to the reference system for extrapersonal space impaired. In our work we detected an interaction between egocentric and allocentric neglect after PA treatment, as graphically represented by the figure 1. Our results, although deserve to be checked with a larger sample of patients with neglect, allowed to measure this interaction.

Point 2: Finally, another minor point concerns the abstract. Authors state: "However, the allocentric component of neglect is frequently impaired, although this is most often in conjunction with the egocentric impairment". What exactly do the authors mean? The fact that the allocentric component is damaged and does not improve after training was already clear before.

Response 2: We thank the reviewer for this suggestion. We modified the phrase in the abstract in order to be more clear.